# Systemic Therapy in Advanced Nodular Melanoma versus Superficial Spreading Melanoma: A Nation-Wide Study of the Dutch Melanoma Treatment Registry

**DOI:** 10.3390/cancers14225694

**Published:** 2022-11-19

**Authors:** Daan Jan Willem Rauwerdink, Remco van Doorn, Jos van der Hage, Alfonsus J. M. Van den Eertwegh, John B. A. G. Haanen, Maureen Aarts, Franchette Berkmortel, Christian U. Blank, Marye J. Boers-Sonderen, Jan Willem B. De Groot, Geke A. P. Hospers, Melissa de Meza, Djura Piersma, Rozemarijn S. Van Rijn, Marion Stevense, Astrid Van der Veldt, Gerard Vreugdenhil, Michel W. J. M. Wouters, Karijn Suijkerbuijk, Monique van der Kooij, Ellen Kapiteijn

**Affiliations:** 1Department of Dermatology, Leiden University Medical Center, Leiden University, Albinusdreef 2, 2300 RC Leiden, The Netherlands; 2Department of Surgery, Leiden University Medical Center, Leiden University, Albinusdreef 2, P.O. Box 9600, 2300 RC Leiden, The Netherlands; 3Department of Medical Oncology, Amsterdam UMC, VU University Medical Center, Cancer Center Amsterdam, De Boelelaan 1118, 1081 HZ Amsterdam, The Netherlands; 4Department of Molecular Oncology & Immunology, Netherlands Cancer Institute, Plesmanlaan 121, 1066 CX Amsterdam, The Netherlands; 5Department of Medical Oncology, GROW School for Oncology and Reproduction, Maastricht University Medical Centre, P. Debyelaan 25, 6229 HX Maastricht, The Netherlands; 6Department of Medical Oncology, Zuyderland Medical Centre Sittard, Dr. H. van der Hoffplein 1, 6162 BG Sittard-Geleen, The Netherlands; 7Department of Medical Oncology & Immunology, Netherlands Cancer Institute, Plesmanlaan 121, 1066 CX Amsterdam, The Netherlands; 8Department of Medical Oncology, Radboud University Medical Centre, Geert Grooteplein Zuid 10, 6525 GA Nijmegen, The Netherlands; 9Isala Oncology Center, Isala, Dokter van Heesweg 2, 8025 AB Zwolle, The Netherlands; 10Department of Medical Oncology, University Medical Centre Groningen, University of Groningen, Hanzeplein 1, 9713 GZ Groningen, The Netherlands; 11Scientific Bureau, Dutch Institute for Clinical Auditing, Rijnsburgerweg 10, 2333 AA Leiden, The Netherlands; 12Department of Surgical Oncology, Netherlands Cancer Institute, Plesmanlaan 121, 1066 CX Amsterdam, The Netherlands; 13Department of Biomedical Data Sciences, Leiden University Medical Centre, Einthovenweg 20, 2333 ZC Leiden, The Netherlands; 14Department of Internal Medicine, Medisch Spectrum Twente, Koningsplein 1, 7512 KZ Enschede, The Netherlands; 15Department of Internal Medicine, Medical Centre Leeuwarden, Henri Dunantweg 2, 8934 AD Leeuwarden, The Netherlands; 16Department of Internal Medicine, Amphia Hospital, Molengracht 21, 4818 CK Breda, The Netherlands; 17Department of Medical Oncology and Radiology & Nuclear Medicine, Erasmus Medical Centre, ‘s-Gravendijkwal 230, 3015 CE Rotterdam, The Netherlands; 18Department of Internal Medicine, Maxima Medical Centre, De Run 4600, 5504 DB Eindhoven, The Netherlands; 19Department of Medical Oncology, University Medical Center Utrecht, Heidelberglaan 100, 3584 CX Utrecht, The Netherlands; 20Department of Medical Oncology, Leiden University Medical Centre, Albinusdreef 2, 2333 ZA Leiden, The Netherlands

**Keywords:** melanoma, immune checkpoint inhibitors, targeted therapy, survival

## Abstract

**Simple Summary:**

Nodular melanoma is associated with a higher locoregional recurrence rate and worse overall survival outcomes. Whether this histologic subtype affects the efficacy of immunotherapy or targeted therapy is unclear. The aim of our multi-center nationwide study is to identify the efficacy of immunotherapy and BRAF/MEKi therapy in metastatic nodular melanoma compared with the efficacy in metastatic superficial spreading melanoma. Our study results demonstrate no difference between the effectiveness of immunotherapy and BRAF/MEKi in metastatic nodular versus superficial melanoma patients. A shorter distant metastasis-free survival and reduced overall survival (measured as the time between primary melanoma up to death or last follow-up) was observed in the nodular melanoma patient group, suggesting worse overal survival of nodular melanoma is mainly driven by propensity of metastatic outgrowth of nodular melanoma after primary diagnosis.

**Abstract:**

Nodular melanoma (NM) is associated with a higher locoregional and distant recurrence rate compared with superficial spreading melanoma (SSM); it is unknown whether the efficacy of systemic therapy is limited. Here, we compare the efficacy of immunotherapy and BRAF/MEK inhibitors (BRAF/MEKi) in advanced NM to SSM. Patients with advanced stage IIIc and stage IV NM and SSM treated with anti-CTLA-4 and/or anti-PD-1, or BRAF/MEKi in the first line, were included from the prospective Dutch Melanoma Treatment Registry. The primary objectives were distant metastasis-free survival (DMFS) and overall survival (OS). In total, 1086 NM and 2246 SSM patients were included. DMFS was significantly shorter for advanced NM patients at 1.9 years (CI 95% 0.7–4.2) compared with SSM patients at 3.1 years (CI 95% 1.3–6.2) (*p* < 0.01). Multivariate survival analysis for immunotherapy and BRAF/MEKi demonstrated a hazard ratio for immunotherapy of 1.0 (CI 95% 0.85–1.17) and BRAF/MEKi of 0.95 (CI 95% 0.81–1.11). A shorter DMFS for NM patients developing advanced disease compared with SSM patients was observed, while no difference was observed in the efficacy of systemic immunotherapy or BRAF/MEKi between NM and SSM patients. Our results suggests that the worse overall survival of NM is mainly driven by propensity of metastatic outgrowth of NM after primary diagnosis.

## 1. Introduction

Cutaneous melanoma is a highly heterogeneous cancer comprised of distinct histologic subtypes based on cell of origin, role of ultraviolet radiation exposure, pattern of oncogenic mutations, and type of histological growth [1,2]. The two major histologic subtypes are superficial spreading melanoma (SSM), covering 70% of the cases, followed by nodular melanoma (NM) with approximately 20% of the cases, whereas the majority of the remaining melanoma cases are of the histologic subtype lentigo maligna melanoma (3–10%) and the histologic subtype acral melanoma is less common [2,3]. It is important to underline the exact histologic subtype of melanoma, as the histologic subtype can potentially play a prognostic role in disease recurrence. NM, in general, has worse prognostic tumor characteristics. including a higher Breslow thickness, ulcerative status, higher dermal mitotic rate, and more frequent satellite lesions [3,4]. The histologic subtype NM is associated with a vertical growth rate and tends to grow more rapidly compared with SSM. As for the mutation profile, NM is more frequently *NRAS* mutated, while SSM harbors the *BRAF* mutation more often. Molecular analysis shows that NM contains a lower mutational load compared with SSM, illustrating the distinct biologic molecular background [5,6,7]. Importantly, primary NM, even corrected for Breslow thickness and ulceration, is associated with lower overall survival and a reduced recurrence-free survival rate compared with primary SSM [8,9,10]. A retrospective study conducted by Lin et al. in melanoma research suggested that the aggressiveness of NM is attributed to a decreased presence of tumor-infiltrating lymphocytes and an upregulation of PD-L1 expression in neoplastic cells compared with SSM; however the exact mechanism of the aggressive behavior of NM has not yet been unraveled.

In the last decade, the advent of immune checkpoint inhibitors and targeted therapy has revolutionized the treatment landscape of metastatic cutaneous melanoma [11,12,13]. The efficacy of immunotherapy ought to be lower in patients with melanoma types with a lower mutation rate, such as acral melanoma, and immunotherapy is more effective in melanoma types with a higher mutation rate, which is the case in the histologic subtype desmoplastic melanoma [14,15]. Despite this, it is unclear whether the primary histologic subtype NM affects the efficacy of immunotherapy and targeted therapy, as the exact significance of the lower mutational profile of NM compared with SSM remains inconclusive.

To date, only two studies compared the efficacy of systemic immune checkpoint inhibitors in NM versus SSM patients and demonstrated contradictory results: Lattanzi et al. observed no difference in survival outcomes of NM versus SSM patients treated with immunotherapy (anti-PD-1 n = 29, anti-CTLA-4 n = 119), while Pala et al. displayed an improved survival of NM patients treated with immunotherapy compared with SSM patients (anti-PD-1 n = 35, anti PD-1/anti-CLTA-4 n = 7) [16,17]. As previously conducted studies were small, unclarity remains regarding the efficacy of immunotherapy and targeted therapy in NM. Identifying the prognostic value of the melanoma subtype can be important in choosing the optimal systemic treatment for the individual patient. Hence, we conducted an analysis using data from a nation-wide prospective registry for systemic treatment of melanoma (the Dutch Melanoma Treatment Registry) to assess survival outcomes of advanced SSM and NM treated with first-line immunotherapy or targeted therapy.

## 2. Methods

### 2.1. Study Design

The Dutch Melanoma Treatment Registry (DMTR) prospectively registers data of systemic therapy in advanced melanoma patients in the Netherlands since 2012 and of resectable stage III and IV melanoma since 2018. This registry and quality assurance has been described in detail by Jochems et al. [18]. The medical ethics committee of each participating hospital approved research using DMTR data and this research was not deemed subject to the Medical Research Involving Human Subjects Act, in compliance with Dutch regulations.

### 2.2. Patients

Eligible patients were 18 years and older, had histologically confirmed advanced (irresectable stage III and IV) cutaneous superficial spreading or nodular melanoma, according to the eighth edition of the American Joint Committee on Cancer (AJCC) classification (including metastases to skin (M1a), lung (M1b), other visceral sites (M1c), and brain (M1d)) [19]. Included patients were naïve to treatment and received first-line systemic anti-CTLA-4 and/or anti-PD-1, or either first-line BRAF inhibitor monotherapy or the combination of BRAF inhibitors and MEK inhibitors. Adjuvant-treated patients were excluded from this study. Data on all patients were collected spanning the period January 2012 to January 2019, while the follow-up data cut off was set at 1 February 2020.

### 2.3. Clinical Variables

Demographic variables (age, gender, and WHO-status) and primary tumor characteristics (Breslow thickness (mm), presence of ulceration, dermal mitosis, satellites, mutation status, location, and histologic subtype were extracted from the DMTR database. Furthermore, clinical data on metastatic melanoma were collected, including site of metastasis, number of disease sites with metastasis, lactate dehydrogenase value (LDH), and details on the type and duration of systemic therapy.

### 2.4. Assessment

A comparative analysis, comparing demographic variables in the NM versus SSM groups based on treatment type, was conducted.

### 2.5. Primary Tumor

Distant metastasis-free survival (DMFS) was determined in both groups and was calculated from the diagnosis of primary melanoma until the occurrence of distant metastasis. Overall survival (OS) was calculated from the diagnosis of the primary tumor until death by any cause or the last moment of follow-up.

### 2.6. Advanced Disease

Progression-free survival (PFS) was calculated from the start of systemic therapy until disease progression or the last moment of follow-up. Furthermore, the response to therapy was assessed and included progressive disease (PD), stable disease (SD), partial response (PR), or complete response (CR). An objective response rate to therapy was calculated per treatment type by comparing the best overall response between NM and SSM patients. Lastly, OS was calculated from the start of systemic therapy until death by any cause or last moment of follow-up and compared between the NM versus SSM groups based on treatment type.

### 2.7. Statistical Analysis

Descriptive analysis was conducted to assess demographic variables, clinicopathological characteristics, and treatment type. Identified frequencies of variables were compared between the SSM and NM groups, using a Chi-square test or Wilcoxon rank test. Survival analyses were conducted with the Kaplan–Meier method and compared with the log rank test across each type of treatment group. Patients not reaching the endpoint were censored at the date of the last contact.

Cox regression analysis was performed to correct for potential confounders. *p*-values were two-sided and *p*-values less than 0.05 were considered to be statistically significant. All statistical analyses were conducted using IBM SPSS Statistics version 24 (Armonk, New York, NY, USA).

## 3. Results

Between 2012 and 2021, a total of 2685 advanced (stadium IIIC or stadium IV) SSM and 1329 NM patients were identified (Table 1). Advanced SSM patients were significantly younger, with a median age of 58 (IQR 47–69) compared with the NM group 63 (IQR 52–72) (*p* < 0.01). Patients with NM had ulceration more often (*p* < 0.01), a higher median Breslow thickness ((3.9 mm (IQR 2.4–6.0) versus 1.9 mm in SSM patients (IQR 1.2–3.3) (*p* < 0.01)), and more frequently had dermal mitoses (*p* < 0.01) and satellite lesions (*p* < 0.01) (Table 2), compared with SSM patients. Considering mutation status, NM harbored *NRAS*-mutations more often (24% versus 16% than SSM patients (*p* < 0.01)), while SSM harbored *BRAF* mutations more frequently (61% compared with 49% in NM patients (*p* < 0.01)).

### 3.1. Distant Metastasis Free Survival between Primary Tumor and Advanced Disease

NM patients had a significantly shorter median DMFS compared with SSM patients when adjusting for Breslow thickness, BRAF-status, mitotic rate, and ulceration, respectively, that is, 1.9 years (95% CI 1.7–2.1) and 3.1 years (95% CI 2.9–3.3) (*p* < 0.01) (Kaplan Meier DMFS analysis, Appendix A and Cox regression DMFS analysis, Appendix A, are displayed in the Appendix A). Overall survival calculated from primary tumor up to decease or last follow-up moment, corrected for age, gender, Breslow thickness, BRAF-status, mitotic rate, and ulceration, demonstrated a median OS of 5.9 years (95% CI 2.7–13) and 8.0 years (95% CI 4.0–16) for NM and SSM, respectively (long-rank test *p* < 0.05).

### 3.2. Immunotherapy in Advanced Disease

A total of 747 advanced NM and 1357 SSM patients received first-line anti-CTLA-4, anti-PD-1 or anti-PD-1/anti-CTLA-4. The specific type of the immunotherapy did not differ significantly between NM and SSM patients (*p* = 0.08) (Table 2).

Immunotherapy-treated NM patients were older with a median age of 67 years (IQR 55–74) versus 64 years (IQR 53–73) (*p* = 0.01) and the majority of NM patients were male, 521 patients (70%) versus 812 (60%) SSM patients (*p* < 0.01). No significant differences were observed between the two groups of patients with brain metastases, with metastases present in three or more organ sites, or with elevated LDH levels. Considering response to immunotherapy, NM and SSM patients had similar objective response rates of 47% and 46%, respectively (Table 3).

Progression-free survival demonstrated a median progressive-free survival of 16.2 months (95% CI 17.3–22.9) for NM patients and 18.1 months for SSM patients (95% CI 14–21) (log-rank test *p* = 0.72) (Kaplan–Meier PFS analysis, Appendix A, and Cox regression PFS, Appendix A, in the Appendix A).

Overall survival analysis, calculated from the start of therapy up to death or last follow-up, showed a median overall survival of 36 months (95% CI 23–49) for NM patients and a median overall survival of 34 months (95% CI 28–41) for SSM patients (log-rank test *p* = 0.53) (Figure 1a).

Cox regression demonstrated that the histologic subtype NM was not associated with decreased survival (HR 0.90 (95% CI 0.76–1.08)) (Table 4). Factors associated with a decreased overall survival since the start of immunotherapy were the presence of brain metastasis (HR 1.05 95% CI 1.01–1.11), elevated LDH levels at the moment of metastasis detection/diagnosis (HR 1.27 (95% CI 1.17–1.38)), and the presence of *NRAS* mutation (HR 1.16 (95% CI 1.05–1.28), while *BRAF* mutation demonstrated a favorable effect with an HR of 0.69 (95% CI 0.58–0.83) (Table 4).

### 3.3. Targeted Therapy in Advanced Disease

In total, 339 advanced NM and 889 SSM patients were treated with BRAF inhibition monotherapy or BRAF/MEK combination therapy. NM patients received BRAF/MEKi combination therapy more frequently compared with SSM patients, 478 (54%) versus 157 (46%), respectively (*p* = 0.02).

NM patients were significantly older with a median age of 64 (IQR 54–73) versus SSM patients with a median age of 60 years (IQR 50–69) (*p* < 0.01). Regarding characteristics of metastatic disease in the two groups, no difference was observed in elevated LDH levels at the moment of metastasis, brain metastasis, and total organ sites with metastatic lesions. As for treatment response, the objective response rate for NM patients was 46%, and 45% for SSM patients. Kaplan–Meier analysis demonstrated a PFS of 7.4 months (95% CI 6.2–8.6) for NM patients, while PFS was 7 months (95% CI 6.3–7.8) in SSM patients (log-rank test *p* = 0.70). Kaplan–Meier analysis for treatment-related overall survival demonstrated a median overall survival of 9.6 months (95% CI 7.9–11.0) and 9.6 months (95% CI 8.5–11.0) for NM and SSM patients, respectively (Figure 1b) (log-rank test *p* = 0.31).

Cox regression analysis showed a hazard ratio of 0.92 (95% CI, 0.78–1.08) for the NM histologic subtype (Table 5). In addition, the presence of brain metastasis (HR 1.08, 95% CI 1.04–1.13), decreased WHO classification (HR 1.08, 95% CI 1.06–1.11), and elevated LDH levels (HR 1.24, 95% CI 1.12–1.32) were associated with a decreased overall survival.

## 4. Discussion

To the best of our knowledge, this is the largest prospective cohort study investigating the efficacy of immune checkpoint inhibitors and targeted therapy in advanced NM compared with SSM patients.

NM patients had a significantly shorter median DMFS compared with SSM patients when adjusting for Breslow thickness, BRAF-status, mitotic rate, and ulceration.

No significant difference in terms of overall survival upon start of systemic therapy was observed in the NM versus SSM group: immune checkpoint inhibition-related survival analysis demonstrated similar survival outcomes. A multivariate analysis, corrected for metastatic and demographic variables, revealed that the histologic subtype NM was not independently associated with decreased treatment-related survival in immunotherapy patients. Considering patients treated with BRAF/MEKi, treatment-related survival analysis showed that survival outcomes did not differ between the NM and SSM group, and multivariate Cox regression analysis demonstrated that the histologic subtype NM was not associated with decreased survival in BRAF/MEKi-treated patients. Interestingly, we did not observe that gender was an independent risk factor for survival in this group, as, is in contrast to the recently published study conducted by Vellano et al. in Nature, it demonstrated that female patients treated with BRAF/MEKi neo-adjuvant treatment had significantly better relapse-free survival rates compared with male patients.

Regarding the importance of histologic subtype of NM in the metastatic setting, only one study, in 21 NM patients, performed by Pala et al., analyzed the survival outcome in addition to the metastatic immunologic behavior of NM compared with SSM and demonstrated a prolonged survival of NM versus SSM [16]. The study attributes the improved survival in NM patients compared with SSM patients to an overexpression of MHC-II molecules and IFN gamma signature, which are both involved in antigen processing and presentation mechanism, which play a significant role in tumor immunogenicity. Despite the improved survival outcome in NM patients, the study was limited in size (anti-PD-1 n = 35, anti-PD-1/anti-CTLA-4 = 7). In contrast, a study conducted by Lantazzi et al. demonstrated no improved survival for metastatic NM compared with SSM treated with immune checkpoint inhibition [17]. Nonetheless, this study was also limited by the fact that the most given treatment was anti-CTLA-4, with only 29 out of the 119 patients receiving anti-PD-1.

In spite of the published results on advanced NM treated with immunotherapy, no large study has been performed investigating the efficacy of targeted therapy. Only the study by Lantazi et al. analyzed the efficacy of targeted therapy in NM and SSM patients and demonstrated a decreased survival for BRAF-mutated NM as compared with BRAF-mutated SSM patients, and histologic subtype NM in the multivariate analysis was independently associated with a decreased survival. However, the power of this study was limited as only 52 patients were included.

It is interesting that a large population-based cross-sectional analysis performed by Allais et al. showed that the diagnosis of primary detected histologic subtype NM, corrected for Breslow thickness and ulceration, was associated with a decreased 5-year relative survival compared with SSM, suggesting that the histologic subtype should be taken into consideration in making treatment decisions [9]. In addition, a large international multi-center study, conducted by Di Carlo and colleagues, demonstrated similar results, with an increased hazard ratio for death in patients with NM (N = 5375) compared with SSM patients (N = 19.592), adjusted for sex, age, and disease stage at diagnosis [8].

The reduced overall survival (measured from primary melanoma up to death), as mentioned in these studies, could be explained by the shorter distant-free metastasis survival for NM versus SSM, as we found in our analysis.

Considering similar treatment-related survival outcomes in advanced SSM and NM patients, we hypothesize that decreased overall survival, measured as time from diagnosis of the primary tumor up to death or the last follow-up moment, in NM patients is mainly driven by primary tumor characteristics and primary tumoral genetic environment, leading to a shorter distant metastasis-free survival. Thus, if NM metastasizes earlier, this will ultimately lead to a worse prognosis. Yet, the histologic subtype NM has not been considered a prognostic metastatic variable, despite a shorter distant metastasis-free survival compared with SSM patients. This underlines the importance of reassessing the follow-up concerning NM patients, and the histologic subtype should be taken into consideration when a decision with regards to adjuvant immunotherapy is made, in order to prolong recurrence-free survival and distant metastasis-free survival.

## 5. Conclusions

Our study shows similar efficacy of immune checkpoint inhibition and BRAF/MEKi in advanced NM compared with SSM patients. However, overall survival, measured as the moment of primary diagnosis up to decease or the last follow-up moment, is shorter because of a shorter distant metastases-free interval in NM as compared with SSM. This might have implications for the follow-up from primary tumor diagnosis and for the consideration of (neo) adjuvant therapy. Future studies should focus on the biologic metastatic behavior.

## Figures and Tables

**Figure 1 cancers-14-05694-f001:**
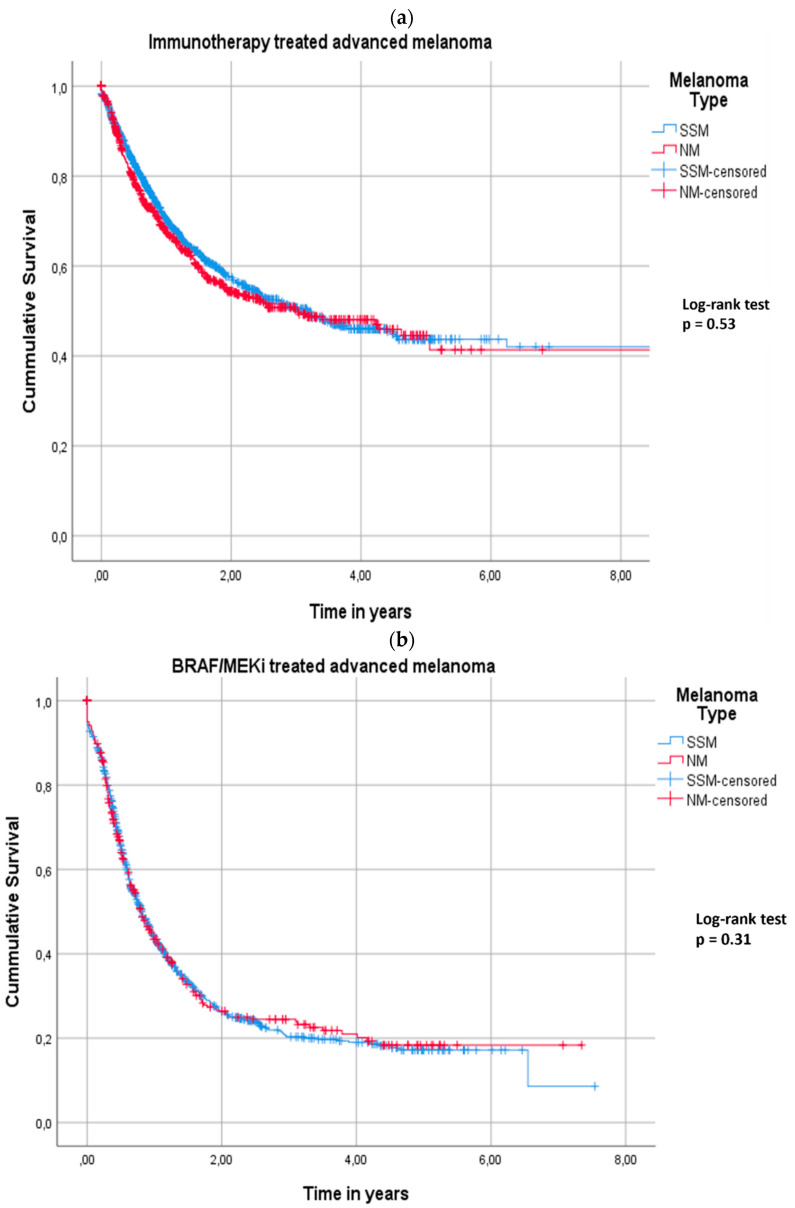
Kaplan–Meier curve survival analysis demonstrating the cumulative survival of NM (red) versus SSM (blue) patients treated with immune checkpoint inhibition (**a**) and Kaplan–Meier curve survival analysis demonstrating the cumulative survival of NM (red) versus SSM (blue) patients treated with immune checkpoint inhibition (**b**).

**Table 1 cancers-14-05694-t001:** The demographic and tumor characteristics of the entire cohort of SSM and NM patients.

Variables	SSM (N = 2685)	NM (N = 1329)	*p*-Value
Median age at moment of diagnosis (IQR)	58 (47–69)	63 (52–72)	
Gender—no. (%)			**<0.01**
Female	1133 (42)	457 (34)	
Male	1552 (58)	872 (66)	
WHO—no. (%)			**0.03**
0	1338 (50)	702 (53)	
1	762 (28)	345 (26)	
>1	290 (11)	125 (10)	
Not reported	288 (11)	157 (12)	
Location primary melanoma—no. (%)			**<0.01**
Head/neck	344 (13)	227 (17)	
Trunk	1405 (52)	589 (44)	
Extremities	903 (34)	497 (37)	
Acral	33 (1)	16 (1)	
Breslow thickness in mm (IQR)	1.9 (1.2–3.3)	3.9 (2.4–6.0)	**<0.01**
Ulceration—no. (%)			**<0.01**
Absent	1602 (60)	580 (44)	
Present	796 (30)	644 (49)	
Unknown	275 (10)	88 (7)	
Dermit—no. (%)			**<0.01**
None	287 (11)	98 (7)	
Any	1391 (52)	820 (62)	
Not reported	978 (36)	400 (30)	
Satellite lesions *			**<0.01**
None	3917 (80)	1748 (77)	
Any	380 (8)	295 (13)	
Not reported	574 (12)	231 (10)	
Mutation status—no. (%) **			
BRAF mutation	1629 (61)	655 (49)	**<0.01**
NRAS mutation	439 (16)	323 (24)	**<0.01**
KIT mutation	24 (0.01)	13 (0.01)	0.08

* Satellite lesions and/or in-transit metastasis. ** Total tested patients tested taken as the denominator.

**Table 2 cancers-14-05694-t002:** Comparative analysis demonstrating demographic, treatment, and metastatic variables in SSM and NM patients treated with first-line systemic immunotherapy. The right side of the table displays a comparative analysis of demographic, treatment, and metastatic variables in SSM and NM patients treated with first-line targeted therapy.

First-Line Systemic Immunotherapy				First-Line Targeted Therapy			
	SSM (N = 1357)	NM (N = 747)	*p*-Value		SSM (N = 889)	NM (N = 339)	*p*-Value
Treatment type			0.08				**0.02**
Anti-CTLA-4	277 (20)	185 (25)		BRAF	411 (46)	182 (54)	
Anti-PD-1	865 (64)	464 (62)		BRAF/MEK	478 (54)	157 (46)	
Anti-PD-1/anti-CTLA-4	215 (16)	98 (13)					
Median age (IQR)	64 (53–73)	65 (55–74)	**0.01**		60 (50–69)	64 (54–73)	**<0.01**
Gender—no. (%)			**<0.01**				0.22
Female	545 (40)	226 (30)			394 (44)	137 (40)	
Male	812 (60)	521 (70)			495 (56)	202 (60)	
WHO—no. (%)			0.22				0.34
0	863 (64)	479 (64)			324 (36)	143 (42)	
1	371 (27)	189 (25)			309 (35)	114 (34)	
>1	49 (4)	30 (4)			167 (19)	48 (14)	
Not reported	73 (5)	49 (7)			89 (10)	34 (10)	
Brain metastasis			0.37				0.99
Not present	1112 (82)	601 (80)			538 (61)	211 (62)	
Present	216 (16)	135 (18)			330 (37)	121 (36)	
*Asymptomatic* ﮺	133	84			112 (34)	44 (36)	
*Symptomatic*	83	51			208 (63)	77 (64)	
Not reported	29 (2)	11 (2)			21 (2)	7 (2)	
LDH			0.06				**0.03**
Normal	1018 (75)	591 (79)			424 (48)	176 (52)	
Elevated	311 (23)	147 (20)			443 (50)	146 (43)	
Not determined	24 (2)	9 (1)			19 (2)	17 (5)	
Organ sites with metastasis			0.35				0.97
<3	442 (33)	236 (32)			44 (5)	18 (5)	
>2	720 (53)	417 (56)			729 (82)	262 (77)	
Unknown	195 (14)	94 (13)			116 (14)	59 (18)	

**Table 3 cancers-14-05694-t003:** Objective response per treatment group stratified per histologic subtype melanoma. The objective response rate is calculated as the sum of complete responses and partial responses.

	Immunotherapy		BRAF/MEK	
	SSM	NM	SSM	NM
Complete response	196 (17)	108 (16)	17 (2)	16 (9)
Partial response	346 (30)	194 (30)	362 (43)	120 (37)
Stable disease	32 (3)	17 (3)	63 (7)	29 (9)
Progressive disease or death	596 (51)	336 (51)	401 (48)	162 (50)
Objective response rate	47%	46%	45%	46%

**Table 4 cancers-14-05694-t004:** Multivariable treatment-related Cox regression analysis in patients treated with immunotherapy. Significant values are highlighted in bold.

Variables		N	Hazard Ratio–95% CI	*p*-Value
Age		2104	1.00 (0.99–1.01)	0.48
Gender	Male	1333	Reference	
	Female	771	0.89 (0.74–1.06)	0.19
WHO	0–1	1902	Reference	
	2–4	79	1.02 (0.99–1.06)	0.20
Treatment type	Anti-CTLA-4	462	Reference	
	Ant-PD-1/Anti-CLTA-4	1642	0.64 (0.53–0.76)	**<0.01**
LDH	Not elevated	1609	Reference	
	Elevated	458	1.27 (1.17–1.38)	**<0.01**
Cerebral disease	Absent	1713	Reference	
	Present	351	1.05 (1.01–1.11)	**0.03**
Total organ sites	<3	678	Reference	
	>2	1137	1.03 (0.87–1.20)	0.76
Melanoma	SSM	1357	Reference	
	NM	747	0.90 0.76–1.08)	0.26
BRAF mutation	Absent	1053	Reference	
	Present	916	0.69 (0.58–0.83)	**<0.01**
NRAS mutation	Absent	1112	Reference	
	Present	574	1.16 (1.05–1.28)	**<0.01**

**Table 5 cancers-14-05694-t005:** Multivariable treatment-related Cox regression analysis in patients treated with targeted therapy. Significant values are highlighted in bold.

Variable		N	Hazard Ratio–95% CI	*p*-Value
Age		1228	1.004 (0.99–1.01)	0.08
Gender	Male	697	Reference	
	Female	531	0.98 (86–1.14)	0.87
WHO				
	0–1	890	Reference	
	2–4	123	1.08 (1.06–1.11)	**<0.01**
Treatment type	BRAF mono therapy	593	Reference	
	BRAF/MEKi	635	0.80 (0.74–0.86)	**<0.01**
LDH	Not elevated	600	Reference	
	Elevated	589	1.24 (1.12–1.32)	**<0.01**
Cerebral disease	Absent	749	Reference	
	Present	451	1.08 (1.04–1.13)	**<0.01**
Total organ sites	<3	62	Reference	
	>2	991	0.94 (0.78–1.13)	0.50
Melanoma	SSM	889	Reference	
	NM	339	0.92 (0.78–1.08)	0.30

## Data Availability

The data presented in this study are available upon request from the corresponding author. The data are not publicly available because of the protection of privacy.

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
