# Peer review of "Systemic Therapy in Advanced Nodular Melanoma versus Superficial Spreading Melanoma: A Nation-Wide Study of the Dutch Melanoma Treatment Registry"

_cancers, 2022, doi:10.3390/cancers14225694_

Round 1
Reviewer 1 Report
The authors compare the efficacy of immunotherapy and BRAF/MEK inhibitors (BRAF/MEKi) in advanced nodular melanoma to superficial spreading melanoma.
The authors should improve some aspects of the manuscript:
Introduction: line 70: The authors say: 'The two major histologic sub-types are superficial spreading melanoma (SSM), covering 70% of the cases, followed by nodular melanoma (NM) with approximately 20% of the cases, whereas acral lentiginous melanomas are less common'. They should say : In third place should be the lentigo maligna melanoma and in last place the acral lentiginous melanoma.
Results:
-The Breslow p-value is missing in Table 1.
-The authors state in: 'Progression free survival demonstrated a median progressive free survival of 18 months (95% CI 14 – 23) for NM patients and 17 months for SSM patients (95% CI 14 – 21) (log-rank test p =0.72)' . Authors should indicate in which table or figure these data appear.
- The authors state in: line 201-203 'Progression free survival demonstrated a median progressive free survival of 18 months (95% CI 14 – 23) for NM patients and 17 months for SSM patients (95% CI 14 – 21) (log- rank test p=0.72)' and in line 179 -180 'NM patients had a significantly shorter median DMFS compared to SSM patients when adjusting for Breslow thickness, BRAF-status, mitotic rate and ulceration, respectively, 1.9 years (CI 95% 1.7 – 2.1) and 3.1 years (CI 95% 2.9 – 3.3) (p<0.01)' . The authors should indicate, in both cases, in which table or figure these data appear. Specifically, in the last of the cases, it is especially relevant, since they are comparing two histological types of melanoma in which there are statistically significant differences in the Breslow index (the adjustment is necessary to compare the data). On the other hand, mitoses do not appear in Table 1.
- On page 8, lines 235-242, they should put 95% CI instead of CI 95% (in any case, always put the 95% confidence interval in the same style)
- Put the values of the log-rank test in figures 1.a and 1.b
References: lack of homogeneity in the number of authors in the references
Author Response
Thank you for the comments. We have adjusted missing P-values in the papers. Also, we have added a supplementary file in which the requested files by the reviewer can be found. The CI are adjusted accordingly.Reviewer 2 Report
I thank the academic editor for giving me the pleasure of reviewing this interesting paper in which the authors conduct a study on the effectiveness of the immunotherapy and BRAF/MEKi in patients suffering of Nodular Melanoma (NM) vs Superficial Spreading type Melanoma (SSM). Overall, in my opinion, this manuscript is well written and conducted, with a few comments about this.
Introduction: The authors explain the reasons to conduct a study like this, highlighiting the “gap” in the literature with only 2 previous study (Lattanzi et al. and Pala et al.) that assessed this topic with different and contrasting results. I suggest to the authors to improve this section with a small paragraph about the histopathological diagnosis of Nodular and SSM because the diagnosis is the key for the treatment. Regarding this, I suggest (from a quick research inPubmed and Medline) these papers:
Tas F, Erturk K. Major Histotypes in Skin Melanoma: Nodular and Acral Lentiginous Melanomas Are Poor Prognostic Factors for Relapse and Survival. Am J Dermatopathol. 2022 Nov 1;44(11):799-805. doi: 10.1097/DAD.0000000000002264. Epub 2022 Jul 19. PMID: 35925149.
Mandalà M, Rutkowski P, Galli F, Patuzzo R, De Giorgi V, Rulli E, Gianatti A, Valeri B, Merelli B, Szumera-Ciećkiewicz A, Massi D, Maurichi A, Teterycz P, Santinami M. Acral lentiginous melanoma histotype predicts outcome in clinical stage I-II melanoma patients: an International multicenter study. ESMO Open. 2022 Jun;7(3):100469. doi: 10.1016/j.esmoop.2022.100469. Epub 2022 Apr 11. PMID: 35421840; PMCID: PMC9271470.
Chang GA, Robinson E, Wiggins JM, Zhang Y, Tadepalli JS, Schafer CN, Darvishian F, Berman RS, Shapiro R, Shao Y, Osman I, Polsky D. Associations between TERT Promoter Mutations and Survival in Superficial Spreading and Nodular Melanomas in a Large Prospective Patient Cohort. J Invest Dermatol. 2022 Oct;142(10):2733-2743.e9. doi: 10.1016/j.jid.2022.03.031. Epub 2022 Apr 22. PMID: 35469904; PMCID: PMC9509439.
Cazzato G, Colagrande A, Cimmino A, Demarco A, Lospalluti L, Arezzo F, Resta L, Ingravallo G. The Great Mime: Three Cases of Melanoma with Carcinoid-Like and Paraganglioma-Like Pattern with Emphasis on Differential Diagnosis. Dermatopathology (Basel). 2021 May 13;8(2):130-134. doi: 10.3390/dermatopathology8020019. PMID: 34068376; PMCID: PMC8161759.
Material and Methods:
Line 106 and 151: please remove the space between the two words.
Results: This section is clear and well written in which authors explain that patients suffering from Nodular Melanoma have more problems in terms of Distance metastasis free-survival (DMFS) and Overall Survival (OS) respect to patients with advanced SSM. The tables are OK.
Discussion: authors discuss the results of our study that demonstrate that in terms of OS there weren’t difference between two groups treated with target therapy and/or BRAF/MEKi.
Please, correct the style of references with one suitable for Cancers.
Author Response
Thank you for the comments. We have adjusted our introduction section in which a broader background of the histologic subtype nodular melanoma is been given. Additionally we have a adjusted the spelling errors.Reviewer 3 Report
An excellent paper, this is the largest prospective cohort study assessing the efficacy of immunotherapy and targetedtherapy in advanced nodular melanoma compared to superficial spreading patients. Material and methods are clearly outlined, inclusion criteria are well defined, statistics are appropriate, results are clearly presented with illustrative tables, discussion is outstanding. It is a landmark paper for worldwide clinicians involved in the treatment of melanoma patients.
Author Response
Thank you for the kind review. We have adjusted our paper: minor errors have been removed.Reviewer 4 Report
This is a well conducted study on large patient numbers which adds important information about the role of histologic subtypes of melanoma in overall survival and response to treatment.
Minor comments relate to recent studies showing influence of female sex in response to targeted therapy Vellano et al Nature and possible influence of sex on response to ICI eg Santoni et al. They could be quoted if even just to dispel
Author Response
Thank you for this most important topic. We have added this into our discussion section.Round 2
Reviewer 2 Report
none